# Pkd2l1 is required for mechanoception in cerebrospinal fluid-contacting neurons and maintenance of spine curvature

Jenna R. Sternberg [1,6], Andrew E. Prendergast[1], Lucie Brosse[2], Yasmine Cantaut-Belarif[1], Olivier Thouvenin [1,3], Adeline Orts-Del'Immagine[1], Laura Castillo[2], Lydia Djenoune[1,7], Shusaku Kurisu[4,8], Jonathan R. McDearmid[5], Pierre-Luc Bardet[1], Claude Boccara[3], Hitoshi Okamoto[4], Patrick Delmas[2] & Claire Wyart[1]

Defects in cerebrospinal fluid (CSF) flow may contribute to idiopathic scoliosis. However, the mechanisms underlying detection of CSF flow in the central canal of the spinal cord are unknown. Here we demonstrate that CSF flows bidirectionally along the antero-posterior axis in the central canal of zebrafish embryos. In the *cfap298*[tm304] mutant, reduction of cilia motility slows transport posteriorly down the central canal and abolishes spontaneous activity of CSF-contacting neurons (CSF-cNs). Loss of the sensory Pkd2l1 channel nearly abolishes CSF-cN calcium activity and single channel opening. Recording from isolated CSF-cNs in vitro, we show that CSF-cNs are mechanosensory and require Pkd2l1 to respond to pressure. Additionally, adult *pkd2l1* mutant zebrafish develop an exaggerated spine curvature, reminiscent of kyphosis in humans. These results indicate that CSF-cNs are mechanosensory cells whose Pkd2l1-driven spontaneous activity reflects CSF flow in vivo. Furthermore, Pkd2l1 in CSF-cNs contributes to maintenance of natural curvature of the spine.

[1] Sorbonne Universités, UPMC Univ Paris 06, Inserm, CNRS, AP-HP, Institut du Cerveau et de la Moelle épinière (ICM) - Hôpital Pitié-Salpêtrière, Boulevard de l'hôpital, F-75013 Paris, France. [2] Aix Marseille Université, CNRS, Laboratoire de Neurosciences Cognitives-UMR7291, F-13344 Marseille, France. [3] PSL Research University, Institut Langevin, ESPCI Paris, F-75005 Paris, France. [4] Riken Brain Science Institute, Wako, Saitama 351-0198, Japan. [5] Department of Neuroscience, Psychology & Behaviour, College of Medicine, Biological Sciences and Psychology, University of Leicester, Leicester LE1 7RH, UK. [6] Present address: Department of Molecular and Cellular Biology, Harvard University, Cambridge, MA 02138, USA. [7] Present address: Department of Medicine, Nephrology Division and Harvard Medical School, Department of Genetics, Massachusetts General Hospital, Charlestown, MA 02115, USA. [8] Present address: Department of Cell Biology, Tokushima University Graduate School of Medical Science, Tokushima 770-8503, Japan. These authors contributed equally: Lucie Brosse, Yasmine Cantaut-Belarif. Correspondence and requests for materials should be addressed to A.E.P. (email: andrew.prendergast@icm-institute.org) or to C.W. (email: claire.wyart@icm-institute.org)

Cerebrospinal fluid (CSF) is secreted by the choroid plexus and circulates within the ventricular system of the brain and in the central canal of the spinal cord[1]. The CSF contains nutrients, extracellular vesicles, peptides, and proteins that can modulate development and function of the nervous system[2–6]. Flow of the CSF distributes these molecules throughout the brain ventricles and central canal of the spinal cord. Motile cilia that line the epithelium surrounding the CSF could contribute to this distribution by shaping the flow[4,7,8].

Multiple lines of evidence from zebrafish indicate that cilia or CSF flow defects influence body axis formation at embryonic stages[9,10] as well as spine morphogenesis in juveniles and adults[11]. In mutants with impaired cilia motility, CSF flow in the brain ventricles is dramatically reduced, and adult mutants exhibit pronounced torsion of the spine similar to clinical presentation of idiopathic scoliosis[1]. Observations linking CSF flow and spine morphogenesis in zebrafish are corroborated by observations from patients with Chiari malformations, in which structural defects in the brain lead to CSF flow defects and scoliosis[12,13]. Active CSF circulation or signaling cues distributed by motile cilia may instruct proper body axis formation, though these mechanisms are currently unknown[14]. Elucidating how physical and biochemical properties of CSF can impact morphogenesis requires an understanding of how CSF flows in the spinal cord and how its physical and chemical properties are detected by local receptor cells lining the central canal.

Primary candidates for sensory cells to detect changes in CSF flow or content are the CSF-contacting neurons (CSF-cNs) that line the brain ventricles and central canal of the spinal cord. Originally characterized in over 100 vertebrate species by Kolmer and Agduhr, these cells share a similar morphology and possess an apical extension that contacts the central canal[15,16]. In zebrafish, mouse, and macaque, CSF-cNs also specifically express the transient receptor potential channel Pkd2l1 (polycystic kidney disease 2-like 1), also known as TRPP3[17,18]. Previous studies showed these cells utilize Pkd2l1 to sense pH changes and respond to bending of the spinal cord during locomotion[19–21]. However, whether the cells can directly direct mechanical perturbation or sense CSF flow has not been proven.

Here we investigate flow of the CSF within the spinal cord, detection of flow by CSF-cNs, and how altered flow can lead to spinal defects. Via full-field OCT and imaging of fluorescent beads, we find that CSF flows bidirectionally in the central canal of the spinal cord. In the zebrafish mutant *cfap298tm304*, previously known as *kurly*, reduction of cilia motility[22] slows transport of particles down the length of central canal. Using calcium imaging, we show that ventral CSF-cNs in this mutant lose spontaneous activity in vivo. In a *pkd2l1* null mutant, in which CSF-cNs lack the sensory channel Pkd2l1, ventral CSF-cNs also lose spontaneous calcium activity. Whole-cell patch clamp recordings further demonstrate that opening of a single channel leads to depolarization that is sufficient to trigger action potentials in wild-type CSF-cNs. Due to the link between disrupted CSF flow and CSF-cN activity, we recorded from isolated zebrafish CSF-cNs to determine whether CSF-cNs respond directly to mechanical pressure. Mechanical stimulation that increases single channel opening in wild-type CSF-cNs abolishes channel activity in *pkd2l1* mutants. Finally, we demonstrate that Pkd2l1 is required for maintenance of a straight spine in adulthood.

## Results

### CSF flow in the central canal of the spinal cord.
We took advantage of the embryonic zebrafish, which is transparent and has easily accessible ventricular structures to study the interactions between the CSF flow, cilia motility, and receptor cells. To examine the dynamics of motile cilia in vivo, we imaged cilia in 24 h post-fertilization (hpf) transgenic zebrafish (*Tg(β-actin:Arl13b-GFP)*)[23] at regularly spaced planes along the dorsoventral axis of the central canal (Fig. 1a, Supplementary Fig. 1, Supplementary Movie 1, and Supplementary Table 1). Cilia exhibited the greatest motility in the ventral portion of the central canal (Fig. 1b)[23]. To visualize fluid flow, we imaged exogenous fluorescent beads with spinning disk confocal microscopy or endogenous particles with full-field optical coherence tomography (FF-OCT, Fig. 1c–i). Both endogenous and exogenous particles showed complex dynamics with local vortices (Supplementary Movies 2, 3). Two-dimensional kymographs of the bead trajectories demonstrated that CSF flow is bidirectional in the central canal (Fig. 1e)[24]. Overall, beads moved anteriorly to posteriorly in the ventral portion of the central canal, whereas beads in the dorsal part of the central canal moved in the opposite direction. (Fig. 1i; bead velocities: ventral: $2.8 \pm 0.24\ \mu m\ s^{-1}$; dorsal: $2.45 \pm 0.34\ \mu m\ s^{-1}$). In order to confirm that the circulation of exogenous beads reflected endogenous flow, we implemented FF-OCT, which allows detection of objects with a refractive index contrast based on interference patterns from backscattered photons (Fig. 1f). At embryonic stages, FF-OCT revealed high levels of endogenous particles in the central canal (Fig. 1g). The displacement of endogenous particles was also bidirectional, though velocities in both ventral and dorsal portions of the central canal were slightly higher than exogenous beads (Fig. 1h, i; ventral: $4.38 \pm 0.39\ \mu m\ s^{-1}$; dorsal: $6.84 \pm 0.31\ \mu m\ s^{-1}$).

### Cilia motility and detection of CSF flow by CSF-cNs.
To assess which aspect of flow required cilia motility, we took advantage of the *cfap298* mutant in which cilia are present but have impaired motility due to defects in recruitment of the outer dynein arms[22]. We assessed flow in *cfap298* mutants by injecting fluorescent beads into the hindbrain ventricle and quantified the transport of beads along the length of the central canal over time. Beads injected in the hindbrain ventricle in the *cfap298* mutant were only able to reach the rostral segments of the central canal, whereas beads in control sibling embryos extended to caudal somites (Fig. 2a–c), demonstrating transport down the central canal was impaired in *cfap298* mutants.

To investigate downstream mechanisms that could transduce CSF flow, we asked how changing the flow modulates activity of potential receptor cells. Primary candidates for sensory cells to detect changes in CSF flow or content are the CSF-contacting neurons (CSF-cNs) that line the brain ventricles and central canal of the spinal cord. Using population calcium imaging in 24–30 hpf paralyzed wild-type embryos, we found that CSF-cNs located near ventral cilia within the central canal were highly active (Fig. 2d and Supplementary Movie 4). To investigate whether CSF-cN activity relied on cilia motility, we monitored CSF-cN calcium activity in *cfap298* homozygous mutants. CSF-cN activity in paralyzed *cfap298* mutants at the same development stage was nearly abolished (Fig. 2d, e and Supplementary Movie 5), suggesting that activity of these cells depends on CSF flow induced by motile cilia. We previously showed that the Pkd2l1 channel was necessary for the response of CSF-cNs to active and passive bending of the tail[19] in vivo. One possible explanation for the spontaneous activity of ventral CSF-cNs could be that these cells are mechanosensory and detect local CSF flow.

### Pkd2l1 is required for spontaneous activity in CSF-cNs.
We generated an antibody for zebrafish Pkd2l1 and observed dense localization of Pkd2l1 in the apical extension of CSF-cNs that contacts the central canal (Fig. 3a), as found in mouse and macaque[17]. Based on the localization of Pkd2l1 in the apical extension,

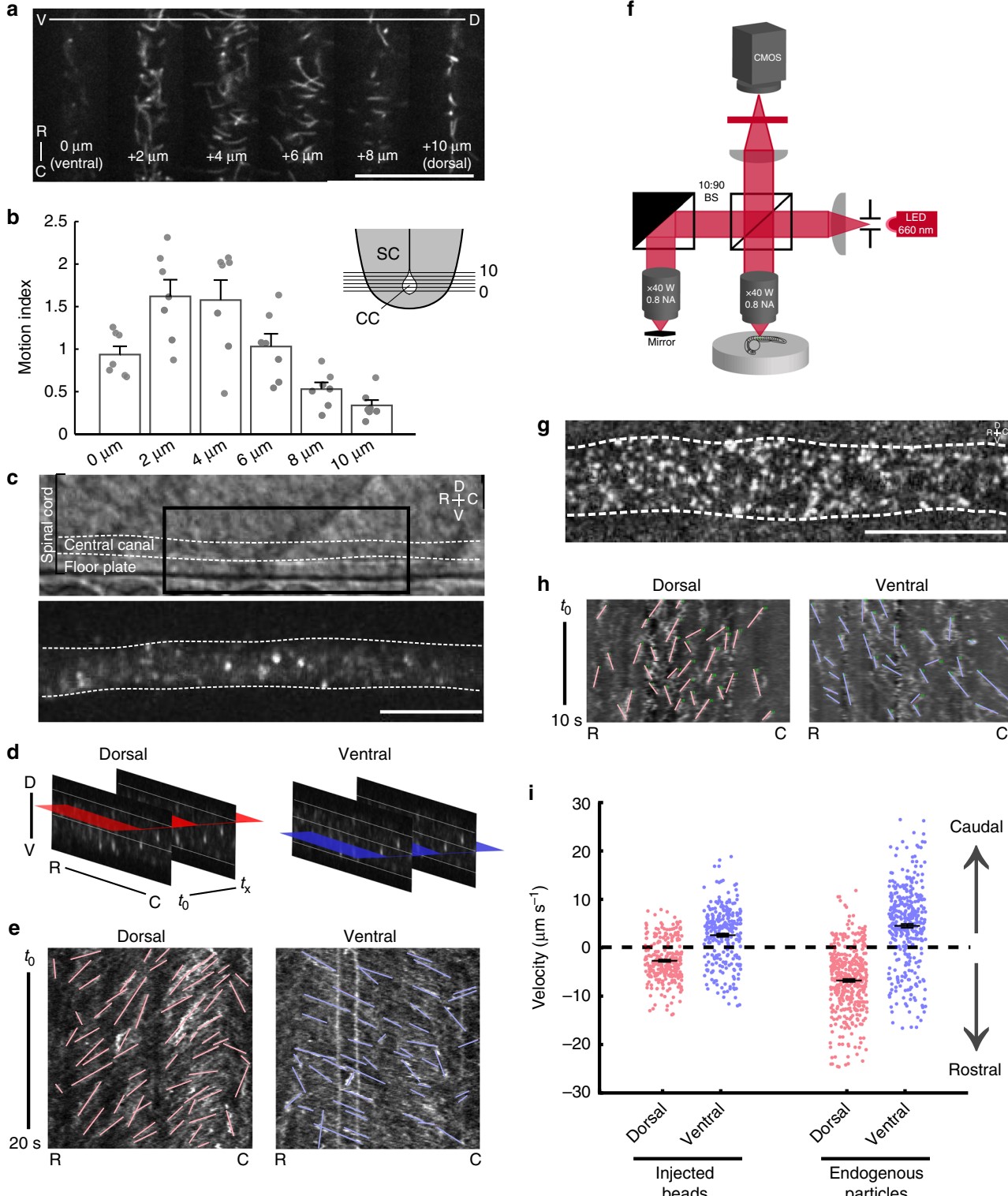

**Fig. 1** Bidirectional flow of cerebrospinal fluid in the central canal of the spinal cord. **a** Single frames from time lapses of 24 h post fertilization (hpf) *Tg(β-actin:Arl13B-GFP)* embryos expressing GFP in cilia. Images taken from progressively dorsal focal planes. Scale: 25 μm. **b** Quantification of ciliary motion from time lapses; ventral planes exhibit more ciliary motion than dorsal planes (one-way ANOVA F = 12.2, p = 5.92 × 10⁻⁷, see Methods for motion index quantification) SC spinal cord, CC central canal. **c** Lateral view of the central canal in transmitted light (top) and filled with fluorescent beads (bottom). Scale: 20 μm. **d** Schematic of horizontal slices taken for kymograph analysis for velocities of beads or endogenous particles. **e** Representative kymographs of beads in the central canal. Analyzed trajectories are in red (dorsal) or blue (ventral). **f** Imaging setup for full-field optical coherence tomography (FF-OCT). **g** Lateral view of the central canal showing endogenous particles obtained from FF-OCT. Scale: 20 μm. **h** Representative kymographs of endogenous particles in the central canal. Analyzed trajectories are in red (dorsal) or blue (ventral). **i** Velocities of exogenous particles (beads) and endogenous particles. Beads: 307 trajectories from n = 7 embryos; endogenous particles: 452 trajectories from n = 7 embryos. Error bars represent s.e.m. D dorsal, V ventral, R rostral, C caudal

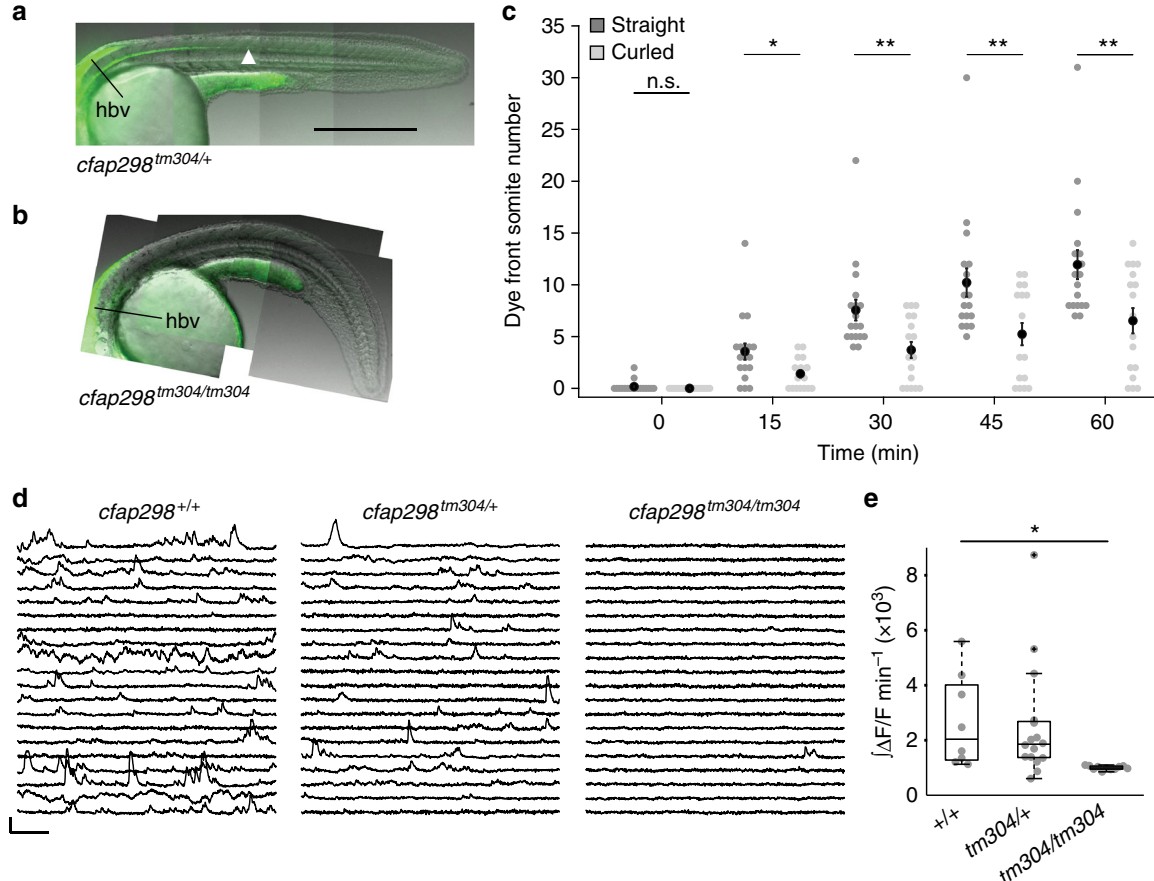

**Fig. 2** cfap298 is required for CSF transport down the rostrocaudal axis and activity of sensory CSF-contacting neurons. Representative images of 28 hpf **a** cfap298[+/+] and **b** cfap298[tm304/tm304] embryos 60 min after injection of fluorescent beads into the hindbrain ventricle (labeled by the black line), the bead front is indicated by an arrowhead in **a**. Scale: 250 μm. **c** Quantification of dye front experiments. ($n = 18$ straight cfap298[+/+] and cfap298[tm304/+] embryos, $n = 17$ curled cfap298[tm304/tm304] embryos). Each point represents one embryo. Error bars represent s.e.m, *$p < 0.05$, **$p < 0.01$. **d** Sample traces of calcium activity in CSF-cNs labeled in Tg(pkd2l1:GCaMP5G) at 25 hpf in cfap298[+/+], cfap298[tm304/+], and cfap298[tm304/tm304]. Traces represent cells with the median integral values taken from all embryos. Scale: 30 s, 30% ΔF/F. **e** Normalized and integrated calcium activity per embryo. Each point represents the average activity across all CSF-contacting neurons for one embryo. cfap298[m304/tm304] embryos ($n = 11$ embryos, 165 cells) had significantly reduced activity compared to wild type ($n = 8$ embryos, 73 cells; one-way ANOVA, $t = 3.7$, $p = 1.7 \times 10^{-3}$, $\alpha = 0.0167$ following Bonferroni correction for multiple testing). The central mark on the box plot indicates the median, the bottom and top edges of the box indicate the 25th and 75th percentiles. The whiskers extend to the most extreme data points not considered outliers, and the outlier is identified with a "+" symbol

CSF-cNs could detect flow or chemical content of the CSF. Immunohistochemistry showed that Pkd2l1 in 1 dpf zebrafish is entirely lost in a pkd2l1 null mutant[19] (pkd2l1[icm02/icm02]; Fig. 3b). We then investigated whether a loss of the Pkd2l1 channel would result in a loss of activity in CSF-cNs. Calcium imaging in paralyzed pkd2l1 mutants revealed a loss of nearly all activity in ventral CSF-cNs, recapitulating the phenotype seen when cilia motility is impaired (Fig. 3c, d and Supplementary Movie 6). To determine the origin of spontaneous activity in CSF-cNs, we made in vivo whole-cell patch clamp recordings. CSF-cNs had a high input resistance ($R_m = 5.9\,\text{G}\Omega \pm 1.9\,\text{G}\Omega$) that enabled extensive single channel opening to be measured in whole-cell mode in vivo (Fig. 3e, f). Spontaneous channel opening was abolished in pkd2l1 mutants (Fig. 3f). Because of the high single channel conductance of Pkd2l1 in CSF-cNs, a single channel opening generated sufficient depolarization to trigger action potentials (Fig. 3g, h).

**Pkd2l1 is required for mechanosensation in CSF-cNs.** Could the Pkd2l1 channel in CSF-cNs directly detect the mechanical stimulus of CSF movement in the central canal? To determine whether CSF-cN are mechanosensory cells, we investigated the properties of isolated CSF-cNs cultured from Tg(pkd2l1:TagRFP) zebrafish larvae (Fig. 4a, b). Cultured CSF-cNs from wild-type embryos exhibited characteristic channel opening comparable to in vivo recordings that were also abolished in pkd2l1 mutants (Fig. 4c). Applying a 4 μm mechanical stimulus transiently on the cell membrane increased the channel opening probability in wild type but not in pkd2l1 mutant CSF-cNs (Fig. 4d–f). Thus, CSF-cNs directly respond to mechanical stimuli in a Pkd2l1-dependent manner, supporting the interpretation that CSF-cN activity reflects CSF flow in vivo.

Together these results suggest that CSF-cNs require the Pkd2l1 channel for detecting CSF flow. An alternative interpretation to the detection of CSF flow by Pkd2l1 could be that Pkd2l1 may be a ciliary protein involved in generating flow. We verified that bidirectional CSF flow was maintained in the pkd2l1 mutant (Supplementary Fig. 2), indicating the loss of calcium activity in pkd2l1 mutants is not a result of a flow defect resulting from the loss of pkd2l1 but of a sensory defect in CSF-cNs.

**Pkd2l1 is required for a straight spine in adulthood.** Disrupted CSF flow due to abnormal cilia motility leads to spinal curvature

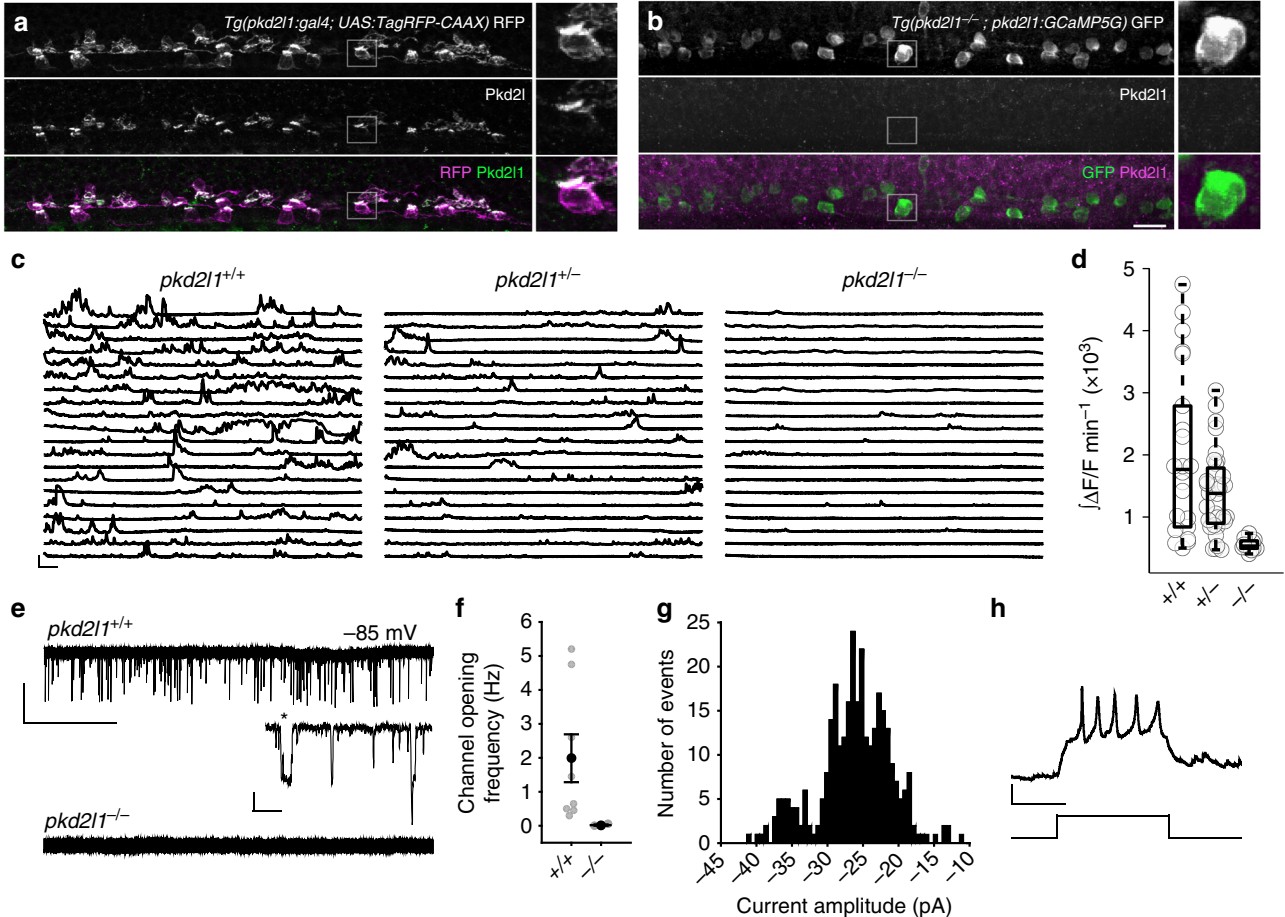

**Fig. 3** Spontaneous activity in CSF-contacting neurons requires Pkd2l1. Immunohistochemistry for zebrafish Pkd2l1 in 30 hpf **a** *Tg(pkd2l1:gal4; UAS:TagRFP-CAAX)* or **b** *Tg(pkd2l1⁻/⁻; pkd2l1:GCaMP5G)* embryos. Right: magnification of the area in the gray box. No Pkd2l1 was detected in *pkd2l1⁻/⁻* embryos. Scale: 20 μm. **c** Calcium imaging traces from ventral CSF-cNs in 24–26 hpf *Tg(pkd2l1:GCaMP5G)* embryos. Traces represent cells with the median integral values taken from all embryos. Scale: 30 s, 100% ΔF/F. **d** Integral quantification of CSF-cN calcium activity. Each point reflects the average calcium activity for all cells from one embryo ($n = 22$ *pkd2l1⁺/⁺*, $n = 32$ *pkd2l1⁺/⁻*, $n = 12$ *pkd2l1⁻/⁻* embryos, $p = 1.94 \times 10^{-13}$, comparison between *pkd2l1⁺/⁺* and *pkd2l1⁻/⁻*, linear mixed models). The central mark on the box plot indicates the median, the bottom and top edges of the box indicate the 25th and 75th percentiles. The whiskers extend to the most extreme data points not considered outliers. **e** Gap-free voltage-clamp (VC) recordings from ventral CSF-cNs show extensive single channel opening exclusively in *pkd2l1⁺/⁺* embryos. Scale: top: 10 s, 20 pA; bottom: 20 ms, 10 pA. **f** Channel opening frequency in ventral CSF-cNs of *pkd2l1⁺/⁺* and *pkd2l1⁻/⁻* embryos ($n = 8$ *pkd2l1⁺/⁺* CSF-cNs from eight embryos, $n = 3$ *pkd2l1⁻/⁻* CSF-cNs from two embryos). Error bars represent s.e.m. **g** Distribution of channel amplitudes at −85 mV in gap-free voltage-clamp recordings of *pkd2l1⁺/⁺* CSF-cNs (mean amplitude 26.0 ± 5.1 pA). **h** Representative firing of a ventral CSF-cN at 28 hpf. Six out of seven ventral CSF-cNs fired single or repetitive action potentials in response to 10–20 pA of injected current. Scale: 100 ms, 20 mV

defects in juvenile and adult zebrafish[11] and is a feature of some spinal curvature defects in patients with Chiari malformations[12,13]. To determine if activity in CSF-cNs contributes to spine morphogenesis, we investigated curvature of *pkd2l1* mutant zebrafish. At larval stages, the Pkd2l1 channel did not contribute to initial formation of the body axis (Supplementary Fig. 3a–c). However, at adult stages, loss of CSF-cN activity in the *pkd2l1* mutant was associated with increased curvature of the spine in the precaudal region, comparable to the thoracic spine in humans. This phenotype resembled human kyphosis, characterized as an abnormally excessive convex curvature of the spine (Fig. 5a–c and Supplementary Fig. 3d–h). Alizarin red staining of bony tissue confirmed that adult *pkd2l1* mutant zebrafish exhibit an abnormal convex curvature of the spine and increased Cobb angle, phenomena consistent with kyphosis in humans (Fig. 5d–g and Supplementary Table 2). Kyphosis was observed in *pkd2l1* mutants compared to control siblings over three generations of fish (Fig. 5 and Supplementary

Fig. 3d–h). Together these results indicate that Pkd2l1-dependent spontaneous activity in CSF-cNs is abolished when cilia motility and CSF flow are lost, suggesting that CSF-cNs detect CSF flow via the Pkd2l1 channel. Sensory functions carried out by Pkd2l1 contribute to the maintenance of spine straightness over time.

## Discussion

Measurements of particle trajectories performed on paralyzed zebrafish embryos show complex bidirectional fluid dynamics in the central canal of the spinal cord. Ventrally, CSF flows from anterior to posterior in the central canal, whereas CSF flows in the reverse direction in the dorsal portions of the central canal. This observation is in consistent with beating cilia concentrated in the ventral central canal (Fig. 1a, b), as previously observed[23]. Using label-free FF-OCT, we found that the central canal has a high density of endogenous particles of high refractive index at embryonic stages. These particles likely comprise exosomes and

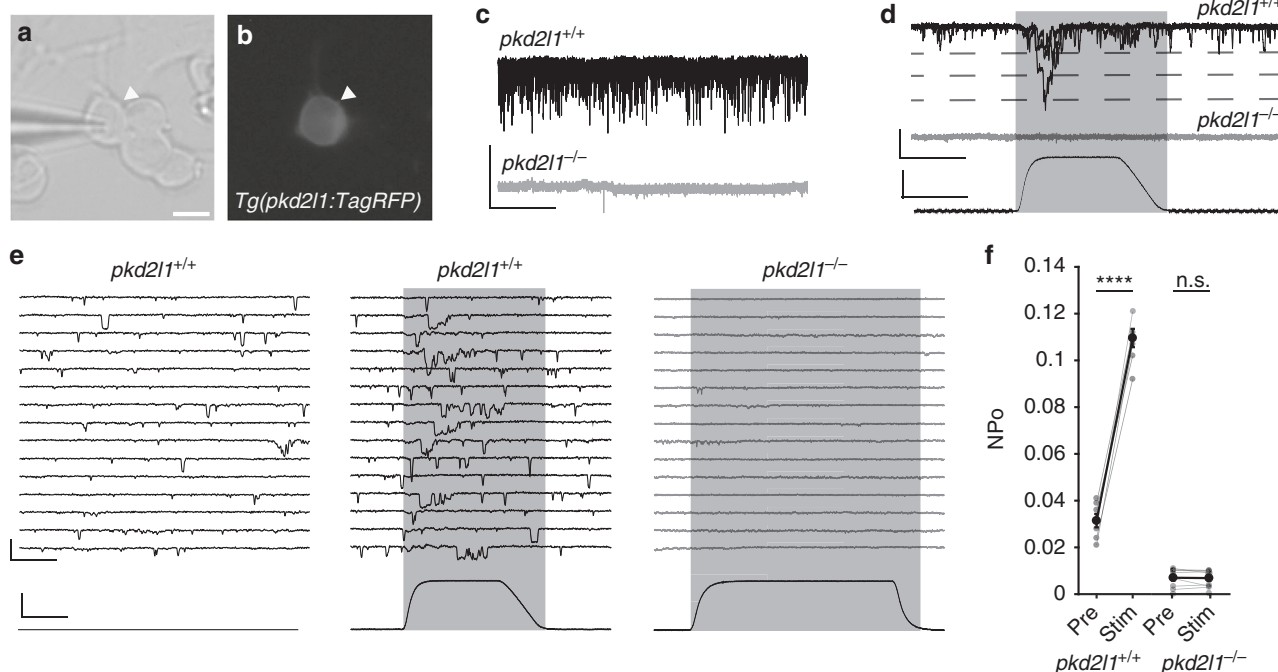

**Fig. 4** CSF-contacting neurons require Pkd2l1 to respond to mechanical pressure. **a** Transmitted light and **b** fluorescent images of a cultured CSF-cN from a *Tg(pkd2l1:TagRFP)* embryo. Scale: 5 μm. **c** Gap-free VC recordings from CSF-cNs in vitro show extensive single channel opening in *pkd2l1+/+* but not *pkd2l1−/−* embryos, comparable to results in vivo. Scale: 2 s, 20 pA. **d** Gap-free VC recording from a cultured CSF-cN while a mechanical stimulus is applied, showing an increase in channel opening during the stimulus in *pkd2l1+/+*. Scale: top: 20 ms, 25 pA, bottom: 20 ms, 2 μm. **e** Gap-free recording from cultured CSF-cNs while a mechanical stimulus is applied. Scale: top: 20 ms, 25 pA, bottom: 20 ms, 2 μm. **f** Quantification of channel opening probability in response to mechanical stimulation of *pkd2l1+/+* and *pkd2l1−/−* CSF-cNs ($n = 7$ *pkd2l1+/+* CSF-cNs, pre vs. stim $p = 1.6 \times 10^{-6}$; $n = 10$ *pkd2l1−/−* CSF-cNs, pre vs. stim $p = 0.76$, paired *t* test). Error bars represent s.e.m.

lipoproteins that carry signaling molecules for development and morphogenesis[25,26]. How do motile cilia contribute to CSF flow in the spinal cord? Transport of exogenous beads down the central canal is less efficient in the *cfap298* mutant that has impaired ciliary motility, consistent with the lack of transport observed in mutants where ciliogenesis was disrupted[8,24]. This result is also consistent with evidence that polarity of motile cilia correlates with flow in rodents[7]. However, the organization of motile cilia in the central canal is highly complex (Supplementary Movie 1), and precise roles of cilia orientation and motility in shaping CSF flow in the spinal cord will require further study.

We demonstrated that CSF-cNs have a direct mechanosensory function that requires the presence of Pkd2l1 in zebrafish. Multiple channels in the PKD family are thought to be involved in mechanosensation[27,28] and cilium-dependent sensing[29,30]. In zebrafish CSF-cNs, the probability of Pkd2l1 channel opening increased with mechanical pressure applied against the membrane. With a small modulation of the Pkd2l1 channel upon mechanosensory stimulation, the unusually high input resistance of CSF-cNs enables these cells to fire action potentials when single Pkd2l1 channels open in zebrafish as in mouse[21].

Pkd2l1 may not act alone in CSF-cN mechanosensory function; differences in the subunit composition of PKD complexes can dramatically affect properties of the channel. CSF-cNs in mouse and zebrafish also express *pkd1l2*[31,32], and Pkd2l1 and Pkd1l2 may form functional heterotetramers trafficked to the cell membrane in the apical extension contacting the central canal. Pkd2l1 in zebrafish CSF-cNs may exist as a homotetramer or as an heterotetramer with other Pkd subunits to trigger CSF-cN spiking, which may confer its specific properties, including the mechanosensory modulation shown here in CSF-cNs. Other ion

channels may also contribute. CSF-cNs in lamprey act as chemoreceptors and possible mechanoreceptors through the activation of an APETx2-sensitive channel[20]. Although this points to a role of ASIC3 channels, ASIC3 orthologs have thus far only been identified in mammalian species[33]. Studies in mouse suggest that CSF-cNs detect changes in pH through ASIC1 channels[21,34]. Additional work is required to understand the specific contribution of these channels to the distinct sensory functions of CSF-cNs.

CSF-cNs are activated when CSF flow is intact and lose activity when cilia motility and transport down the central canal are impaired. One concern is that reduced activity in CSF-cNs results from deformations in the body axis. Another explanation could be that the *cfap298* mutation could affect sensory properties of CSF-cNs themselves because they have a motile cilium[19]. Sensory properties of CSF-cNs are not abolished by *cfap298* disruption as *cfap298* mutants still responded to spontaneous muscle contractions, though the amplitude of these responses was decreased (Supplementary Fig. 4). In *cfap298* mutants, Pkd2l1 is correctly localized to the apical extension as in wild-type zebrafish (Supplementary Fig. 5). However, the decreased sensory response observed in *cfap298* during spontaneous contractions suggests that the motile cilium may contribute to the sensory function of CSF-cNs.

This combination of in vitro and in vivo data point toward CSF-cNs being mechanosensory cells that interface with the CSF and detect CSF flow via the Pkd2l1 channel. Addressing in vivo whether CSF-cNs are responding directly to mechanical components of CSF flow is not possible. Therefore, we cannot eliminate the possibility that chemical signals contribute to CSF-cN sensory activation in vivo. Non-neuronal epithelial cells are capable of releasing compounds in

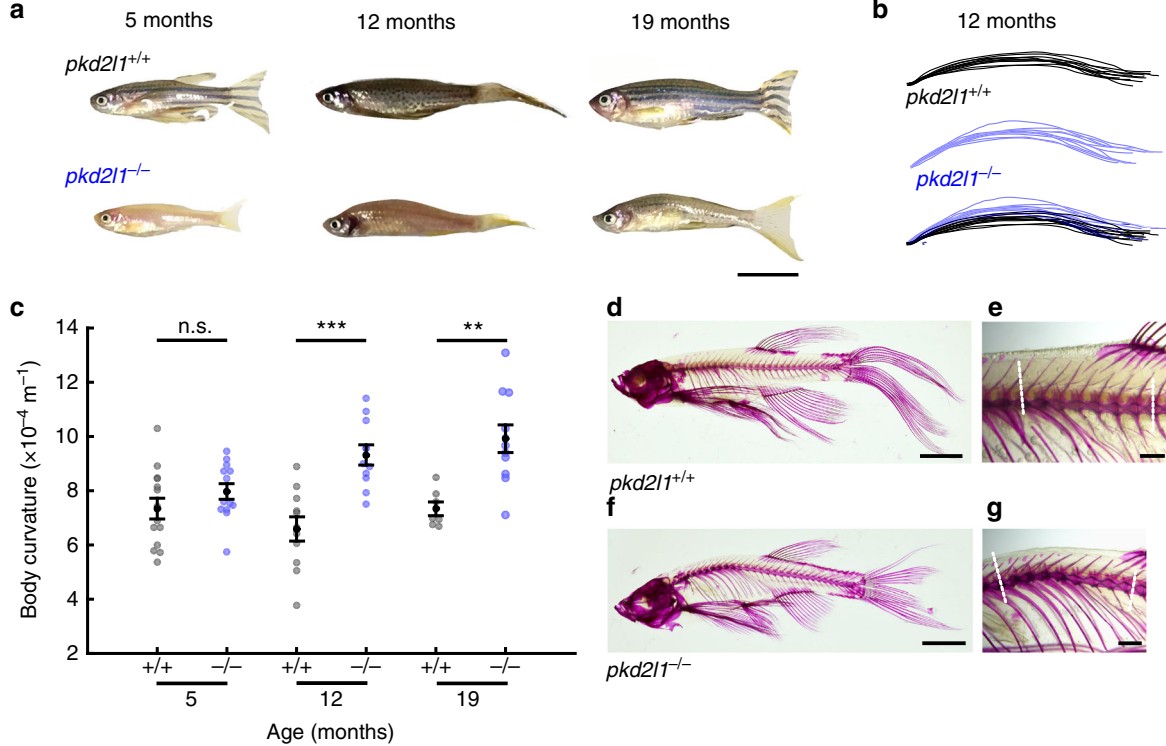

**Fig. 5** Pkd2l1 is necessary to maintain a straight spine. **a** Representative $pkd2l1^{+/+}$ and sibling $pkd21^{-/-}$ adult zebrafish at adult stages. Scale: 1 cm. **b** Body curvature traces from $pkd2l1^{+/+}$ (black) and $pkd2l1^{-/-}$ (blue) adults at 12 months ($n = 11$ $pkd21^{+/+}$, $n = 11$ $pkd2l1^{-/-}$). **c** Quantification of body axis curvature. Each point represents one fish. Error bars represent s.e.m. (5 months: $n = 14$ $pkd2l1^{+/+}$, $n = 13$ $pkd2l1^{-/-}$, $p = 0.02$; 12 months: $n = 11$ $pkd2l1^{+/+}$, $n = 11$ $pkd2l1^{-/-}$, $p = 1.42 \times 10^{-4}$; 19 months: $n = 7$ $pkd2l1^{+/+}$, $n = 11$ $pkd2l1^{-/-}$, $p = 1.5 \times 10^{-3}$; two-sample $t$ test). **d** Alizarin red staining of bones of an adult $pkd2l1^{+/+}$ fish at 20 months of age, showing little to no spinal kyphosis. Scale: 5 mm. **e** High-magnification image of the precaudal region of the $pkd2l1^{+/+}$ spine. Dotted lines indicate vectors used to calculate the Cobb angle. Scale: 1 mm. **f** Alizarin red staining of an adult $pkd2l1^{-/-}$ fish. Note the kyphosis in the precaudal region. Scale: 5 mm. **g** High-magnification image reveals pronounced kyphosis, no torsion is evident (see Supplementary Table 2 for Cobb angles). Scale: 1 mm

response to mechanical stimulation[35]. Chemicals transported down the central canal or local ciliary motility may activate epithelial radial glia that in turn could lead to a release of compounds that activates CSF-cNs[36].

The *pkd2l1* mutation lead to an excessive convex curvature of the spine in three generations of adult zebrafish. Interestingly, *pkd2l1* mutants exhibit an exaggerated Cobb angle at one location in the precaudal spine, a hallmark of kyphosis in humans. Mutations in motile cilia lead instead to strong torsional defects in adult zebrafish that resemble scoliosis[11]. As ciliary mutations and mutations in the Pkd2l1 lead to distinct spine defects, CSF-cN detection of CSF flow appears to contribute to some but not all CSF-dependent defects in spine morphogenesis.

We can only speculate on the mechanisms underlying the contribution of Pkd2l1 in CSF-cNs to body axis maintenance. CSF-cNs in the spinal cord express neuropeptides and mono-amines and modulate excitability of motor circuits via GABAergic synapses[37–39]. As CSF-cNs synapse onto premotor and motor neurons[40–42], decreased CSF-cN activity throughout life may lead to changes in swimming output. Subsequent alterations in biomechanical forces from the water could lead to a difference in the mechanical strain experienced by the fish over time and contribute to a progressive defect. Spinal CSF-cNs also require Pkd2l1 to respond to bending of the body axis and exert postural control during swimming[19,41]. The sensory feedback provided by detection of longitudinal and lateral bending of the tail could influence the shape of the spine. Another possibility is that CSF-cNs may regulate responses to inflammatory molecules in the CSF, as bacterial infection leads

to spinal defects similar to those seen in adult *pkd2l1* mutants[43]. How Pkd2l1-dependent activity in CSF-cNs contributes to defects in spine morphogenesis remains an area for future investigation.

## Methods
**Zebrafish husbandry.** All procedures were approved by the Institut du Cerveau et de la Moelle épinière (ICM) and the National Ethics Committee based on E.U. legislation. All experiments were performed on *Danio rerio* of AB and TL background. For some experiments, $mitfa^{+/-}$ or $mitfa^{-/-}$ animals were used. Embryos were raised in an incubator at 28.5 °C under a 14/10 light/dark cycle until the start of experimentation and staged according to ref. [44].

**Analysis of ciliary movement in the central canal of the spinal cord.** Twenty-four hpf $Tg(\beta\text{-}actin{:}Arl13B\text{-}GFP)$ embryos were mounted dorsal side up in 1.5% agarose and paralyzed with ~2 nL of 500 μM α-bungarotoxin injected into caudal lateral muscle. A spinning disk confocal microscope (3i, Intelligent Imaging Systems) was used to focus on a single optical section at the ventral-most extent of the central canal and a 15 s time-lapse was acquired at 33 Hz imaging frequency. Subsequent time lapses were acquired at 2 μm steps moving from the ventral to dorsal limits of the central canal (see schematic in Fig. 1b and Supplementary Fig. 1).

To analyze ciliary motion, a composite time-lapse was assembled by conjoining each time-lapse side-by-side into a 500-frame movie containing all dorsoventral planes (Fig.1a, Supplementary Fig. 1, and Supplementary Movie 1). Two projections in the time dimension were generated from these composite time lapses, a standard deviation projection (representing variation in pixel intensity, an estimate of ciliary motion) and an average intensity projection (representing overall brightness and labeling intensity). An intensity profile of each projection was then acquired in the *y*-axis with Fiji[45], producing an intensity peak at each focal plane corresponding to standard deviation or average intensity (Supplementary Fig. 1).

Because fluorescence intensity changes depending on transgene copy number and fluctuations in imaging conditions, it was necessary to normalize the standard

deviation plot by dividing it by the average intensity plot to yield an intensity-corrected motion index. Finally, a background subtraction was carried out on the normalized plots (by subtracting the minimum intensity value in each focal plane) and the integral was approximated using the *trapz* function in MATLAB (The Mathworks, Inc.) to yield the values displayed in Fig. 1b. These values represent a rough index of overall pixel intensity fluctuation at each focal plane, and consequently, a proxy measure for ciliary movement.

**Fluorescent bead tracking.** Embryos were manually dechorionated then embedded in 1.5% low-melting point agarose at 24–26 hpf and paralyzed by injection of ~2 nL of 500 μM α-bungarotoxin in the caudal musculature of the tail. Yellow-green 0.02 μm fluorescent FluoSpheres (ThermoFisher Scientific, A8888) were diluted to a 2% concentration by volume in artificial CSF (aCSF, concentrations in mM: 134 NaCl, 2.9 KCl, 1.2 MgCl$_2$, 10 HEPES, 10 glucose, 2 CaCl$_2$; 290 ± 3 mOsm/L, pH adjusted to 7.8 with NaOH) then briefly sonicated. 1–3 nL of the solution were pressure injected in the hindbrain ventricle. Images were acquired using spinning disk confocal (3i Intelligent Imaging Innovations, Inc.) for 20 s at 10 Hz using Slidebook software (3i Intelligent Imaging Innovations, Inc.). Motion of beads or endogenous particles were analyzed in Fiji[45]. Videos were rotated and cropped to reduce the field of view to the central canal. Images were resliced in Fiji, then a 1.5–1.7-μm-wide region at 30 or 85% of the dorsoventral axis was projected as a kymograph to obtain time trajectories of individual beads. Custom MATLAB code was used to obtain start and end points for individual beads and calculate velocities.

**Full-field optical coherence tomography.** The FF-OCT path is based on a Linnik interference microscope configuration illuminated by a temporally and spatially incoherent light source. A high power 660 nm LED (Thorlabs M660L3, spectral bandwidth 20 nm) provided illumination in a pseudo Köhler configuration. A 90:10 beam splitter separates the light into sample and reference arms. Each arm contains a ×40 water immersion objective (Nikon CFI APO ×40 water NIR objective, 0.8 NA, 3.5 mm working distance). In the reference arm, the light is focused onto a flat silicon wafer with a reflection coefficient of about 23.5% at the interface with water. FF-OCT detects any structure that reflects or backscatters light within the sample arm. The backscattered and reflected coefficients depend on the refractive index, size, and shape of the imaged structures. Light returning from both arms is recombined by the entrance beam splitter. The two beams interfere only if the optical path length difference between both arms remains within the coherence length of the system, ensuring efficient optical sectioning. A 25-cm focal length achromatic doublet focuses the light to a high speed and high full well capacity CMOS camera (Adimec, custom built). The overall magnification of the FF-OCT path is ×50. The measured transverse and axial resolutions were 0.525 μm and 4 μm, respectively. Camera exposure was 9.8 ms, and images were acquired at 100 Hz. Direct images are used to reduce the mechanical vibrations caused by movement of the piezo actuator. Due to the low coherence of the setup, it is only sensitive to intensity changes that happen at a given depth inside the coherence gate of the microscope. We acquired sequences of consecutive direct images and computed the standard deviation on groups of images to cancel the incoherent light that does not produce interference. The image obtained therefore corresponds to intensity fluctuations occurring at a given depth, similar to what is obtained by modulating the piezo position. This method allows detection of phase fluctuations caused by small-particle displacement occurring in the 10–100 Hz range. The full-field optical coherence tomography (FF-OCT) setup used is described in additional detail in a separate publication[46]. Twenty-three trials of 10 s duration were obtained from seven embryos between 25 and 32 hpf. Ten-second time series were acquired at 100 Hz at a position around the central canal, identified using GFP fluorescence-labeled cilia in the *Tg(β-actin:Arl13B-GFP)* or *Tg(pkd2l1:GCaMP5G)* transgenic lines.

**Analysis of macroscopic bead flow.** Beads were injected into the hindbrain ventricle as described above. Only embryos in which the beads reached the central canal from the ventricle were selected for analysis for both genotypes. To develop an unbiased method to determine the position of the end of the dye front, regions of interest (ROI) were first defined for each somite in the field of view using a transmitted light image. A plot profile was then generated in the green channel for each somite ROI. If the central canal had been infiltrated by the fluorescent beads at this rostrocaudal somite position, a distinct peak would be apparent in this plot profile. To determine the maximum extent of the dye front in an unbiased fashion, a ratio of the peak central canal fluorescence intensity was divided by the green channel fluorescence adjacent to the central canal. A cutoff value of 1.2 was used to define a somite as infiltrated vs. not infiltrated.

**Immunohistochemistry.** Thirty hpf embryos were killed using 0.2% tricaine, and fixed in 4% paraformaldehyde for 4 h at 4 °C. After three washes in 1× PBS, embryos were blocked in PBS 0.7% Triton, 1% DMSO, 10% normal goat serum, and 1 mg/mL bovine serum albumin overnight at 4 °C. The following primary antibodies were then incubated overnight at 4 °C using the following concentrations in blocking buffer: anti-Pkd2l1 (rabbit, polyclonal, lab-made, 1:200), anti-RFP (mouse, Thermo Fischer Scientific, MA515257, 1:500), and anti-GFP (chicken,

Abcam, ab13970, 1:500). Washes were performed in 1% DMSO and 0.3% Triton-PBS, and secondary antibodies were incubated for 2.5 h at room temperature. All secondary antibodies (Invitrogen, Alexa Fluor-488 goat anti-chicken IgG A21202, Alexa Fluor-488 donkey anti-rabbit IgG A21206, Alexa Fluor-568 goat anti-rabbit IgG A11011, Alexa Fluor-488 goat anti-mouse A11004) were used at 1:500 dilution in blocking buffer.

**Calcium imaging.** Embryos were staged to 24–26 hpf, dechorionated manually, then embedded in 1.5% low-melting point agarose. All imaging was performed following injection in caudal axial muscle with ~2 nL of 500 μM α-bungarotoxin (Tocris Bioscience). Calcium imaging was performed at 4 or 5 Hz with a spinning disk confocal (3i Intelligent Imaging Innovations, Inc.,) for 4 or 8 min. Images were acquired using Slidebook software (3i Intelligent Imaging Innovations, Inc.) and reconstructed online using Fiji[45]. ROIs were manually selected based on a standard deviation Z-projection and ventromedial position in the spinal cord. ΔF/F and normalized integral of activity were calculated with custom scripts written in MATLAB.

**In vivo patch clamp recordings.** Whole-cell recordings were performed in 25–30 hpf *Tg(pkd2l1:gal4; UAS:mCherry)* or *Tg(olig2:DsRed2)*[47] embryos in aCSF. Embryos were pinned through the notochord with 0.025 mm tungsten pins. Skin and muscle from two to four segments between segments five and twelve were dissected using a glass suction pipette. A MultiClamp 700B amplifier, a Digidata series 1440 A Digitizer, and pClamp 10.3 software (Axon Instruments) were used for acquisition. Raw signals were acquired at 50 kHz and low-pass filtered at 10 kHz. Patch pipettes (1B150F-4, WPI) with a tip resistance of 7–9 MΩ were filled with internal solution (concentrations in mM: K-gluconate 115, KCl 15, MgCl$_2$ 2, Mg-ATP 4, HEPES-free acid 10, EGTA 5 or 10, 290 mOsm/L, pH adjusted to 7.2 with KOH with Alexa 488 at 10 μM final concentration). Holding potential was −85 mV, away from the calculated chloride reversal potential (−51 mV). Analysis of electrophysiological data was performed offline using Clampex 10 software (Molecular Devices). Single channel events were identified using a threshold search in Clampfit (Molecular Devices), with a threshold triggered at −8 pA from the baseline and a rejection at −30 pA. Only events lasting longer than 1.5 ms were included for analysis. For CSF-cNs in *pkd2l1*$^{+/+}$ embryos, a 20-s window was used to identify channel events from a gap-free voltage-clamp recording from the first 2–5 min of recording.

**Isolation and dissociation of spinal cord zebrafish embryos.** *Tg(pkd2l1: TagRFP)*-positive zebrafish embryos were then placed in Hank's balanced salt solution (HBSS, 0.137 M NaCl, 5.4 mM KCl, 0.25 mM Na$_2$HPO$_4$, 0.1 g glucose, 0.44 mM KH$_2$PO$_4$, 1.3 mM CaCl$_2$, 1.0 mM MgSO$_4$, 4.2 mM NaHCO$_3$) and anesthetized with 0.02% tricaine. The chorion, yolk, head, and caudal tail were removed. Dissected embryos were then placed into 1 mL HBSS, which was replaced by collagenase type IA solution (2 mg/mL) once the spinal cords were dissected and incubated for 45 min at 37 °C with trituration occurring half time using a P1000 tip. Dissociated tissue was centrifuged for 3 min at 1000 rpm. The supernatant was removed and the pellet resuspended in 1 mL HBSS. The solution was then passed through a 40 μm filter.

**Plating of dissociated zebrafish spinal cord cells.** Dissociated spinal cord cells from 4 dpf *Tg(pkd2l1:TagRFP)*-positive zebrafish were plated on top of spinal cord cells cultured from 2 dpf wild-type zebrafish (to improve adherence) on 12 mm Corning Bioboat Coverslips precoated with Laminine and Poly-Lysine. Cells were then cultured in Dulbecco's modified Eagle's medium (Invitrogen) supplemented with 10% fetal bovine serum, 50 units/mL penicillin–streptomycin, 25 mM glucose, 2 mM L-glutamine, 25 ng/mL NGF, and 4 ng/mL GDNF and incubated overnight at 37 °C in 5% CO$_2$.

**In vitro patch clamp recording and mechanical stimulation.** *Tg(pkd2l1: TagRFP)*-positive CSF-cNs were patch clamp recorded within 1 day of plating. Recordings were performed using borosilicate electrodes (Harvard Apparatus) having a resistance of 5–8 MΩ when filled with a solution containing (mM): 130 KCl, 10 HEPES, 4 CaCl$_2$, 1 MgCl$_2$, and 10 EGTA, 4 Mg-ATP and 0.4 Na-GTP (pH adjusted to 7.4 with KOH, ~300 mOsm/L). The extracellular solution consisted of (mM): 140 NaCl, 10 HEPES, 10 glucose, 1 KCl, 1 MgCl$_2$, and 2.5 CaCl$_2$ (pH adjusted to 7.4 with NaOH, 300 mOsm/L). Whole-cell recordings were made at 24 °C using an Axopatch 200B amplifier (Axon Instruments), filtered at 1–2 kHz, and digitally sampled at 20 kHz. Voltage errors were minimized using 75–80% series resistance compensation. Data acquisition was performed with pClamp 10.2.

Piezoelectrically driven mechanical stimulation was used to apply focal force onto the neurons[48]. Mechanical stimulation was achieved using a fire-polished glass micropipette positioned at 65° from the horizontal plane and cemented to a piezo-electric actuator (Step Driver PZ-100; Burleigh). Voltage-clamped CSF-cNs were mechanically stimulated by the displacement of the probe toward the selected neuron using increments of 0.2–0.5 μm. The probe had a velocity of 800 μm/s during the ramp segment of the command for forward motion. Raw data were analyzed using paired *t* test or Mann–Whitney test depending on the experimental design. Analysis used a combination of Clampfit 10.2 (Molecular Devices), Origin

7.0 (OriginLab), and PRISM 4.0 (GraphPad) softwares. All values are shown as mean ± standard error of the mean (s.e.m.) and *n* represents the number of cells examined.

**Body axis analysis**. Male and female adult zebrafish aged 5, 12, or 19 months were briefly anesthetized in 0.02% tricaine and lateral images were obtained. Curvature was calculated in MATLAB using the LineCurvature2D function. Curvature analysis was performed blinded to fish genotype. *pkd2l1*[+/+] siblings were used as controls. For analysis of 2 dpf *pkd2l1* mutant embryos, the angle of the tail was calculated based on a point placed midway through the eyes, the swim bladder, and on the tip of the tail.

**Alizarin red staining**. Male and female adult *pkd2l1*[+/+] and *pkd2l1*[−/−] fish were killed at 20 months of age in 0.1% tricaine and subsequently fixed in 4% paraformaldehyde for 1 h at room temperature. Fish were then eviscerated, cleared, and stained using 1 mg/mL alizarin red as described in ref. [49]. After staining, fish were photographed using a Nikon digital single-lens reflex camera fitted with an AF-S Micro Nikkor 60 mmF/2.8G ED lens.

**CSF-cN response to muscle contraction in *cfap298*[tm304]**. Unparalyzed 24–30 hpf *Tg(pkd2l1:GCaMP5G)* zebrafish embryos were pinned through the notochord with 0.025 mm tungsten pins and bathed in aCSF. Images were acquired with a 488 nm laser on a spinning disk confocal (3i Intelligent Imaging Innovations, Inc.) at 4 Hz, as performed for calcium activity experiments in paralyzed embryos. Offline analysis to determine contraction amplitude was performed in MATLAB. In order to estimate the timepoint of motion artifacts, the average fluorescence signal for all ROIs was calculated to obtain a single time series. The threshold for motion artifacts was set to be when the absolute value of the time derivative was greater than three times the standard deviation of the single time series. In order to prevent increases in calcium activity being detected as a motion artifact, a filter was applied to impose the motion artifact values to be below the median value of the signal.

## Data availability

Data and MATLAB scripts used for analysis are available from the authors upon request.

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

## Acknowledgements

We thank Prof. Brian Ciruna for the *Tg(β-actin:Arl13b-GFP)* transgenic line and Prof. Rebecca Burdine for the *cfap298^tm304* mutant line, Sophie Nunes-Figueiredo, Natalia Maties, Bogdan Buzurin, and Monica Dicu for fish care, Steven Knafo for advice on human spinal deformities, Vincent Guillemot amd François-Xavier Lejeune for statistical analysis, Annick Prigent for assistance with Alizarin red staining, the histology and imaging ICM-Quant platforms of the ICM, and members of the Wyart Lab for critical feedback (https://wyartlab.org). This work was supported by an ERC Starting Grant "Optoloco" #311673, the HFSP Program Grants #RGP0063/2014 and #RGP0063/2018, the Michelin Corporate Foundation, Mr. Pierre Belle, and a BBSRC program grant BB/N010140/1 to J.R.M. J.R.S. was supported by the Ecole des Neurosciences de Paris. A.E.P. was supported by an EMBO long-term fellowship (ALTF 549–2013) and a Research in Paris grant from the Mairie de Paris.

## Author contributions

J.R.S. and A.E.P., P.D., and C.W. designed the experiments. J.R.S., A.E.P., Y.C.-B., O.T., A.O.-D., L.B., L.C., and L.D. performed the experiments. J.R.S., A.E.P., O.T., P.D., and C.W. analyzed the data. S.K. and H.O. provided the ventricle injection protocol for beads. C.W., P.D., P.-L.B., and C.B. supervised the research. J.R.M. and P.D. provided essential technical training and supervised the research. J.R.S. and C.W. wrote the article with input from all authors.

## Additional information

**Competing interests:** The authors declare no competing interests.

