## [Peer Review File · Nature Communications]

Reviewers' comments:

Reviewer #2 (Remarks to the Author):

In this study, using zebrafish as a model, Sternberg et al. first described the bidirectional flow of cerebrospinal fluid (CSF) in the spinal canal, and showed that the Kurly mutant with affected cilia motility decreased CSF flow and abolished the activity of CSF-contacting neurons (CSF-cNs). Then they demonstrated that the Pkd2/1 channel is required for CSF-cNs to respond to mechanical pressure. Finally, they showed Pkd2/1 mutation led to spinal curvature defects in adult fish. In summary, this study shows that CSF-cNs transduce CSF flow into neural activity via Pkd2/1 channels and indicates that such process is important for natural curvature of the spinal cord.

Overall the experiments were well designed and performed. Considering the function of Pkd2/1 channels in spinal curvature defects may have implications for the treatment of idiopathic scoliosis, I think this work is important and would like to support its publication in Nature Communication after addressing the following concerns.

1. The defect of spinal curvature in the Kurly mutant looks more serious than that in the Pkd2/1 mutant. Therefore, I think the authors should measure and compare the spinal curvature defects in both the Kurly and Pkd2/1 mutant to clarify the importance of Pkd2/1 channels in spinal curvature.
2. In the study, the expression of Pkd2/1 channels was showed in in the apical extension of CSF-cNs in the trunk. Is there any expression of Pkd2/1 channels in the brain. It would be nice to exclude the possibility that spinal curvature defects in Pkd2/1 mutants are not due to the disruption of the activity of neurons in the brain. Therefore, I suggest the authors could express a neurotoxin, like the BoTxBLC, specifically in CSF-cNs to demonstrate their importance in spinal curvature defects.
3. In the fig.S2, the mutant rather than morphant of Kurly should be examined to demonstrate the integrity of CSF-cNs' ability for responding to mechanical stimuli.

Reviewer #3 (Remarks to the Author):

In this Manuscript, Jenna Sternberg et al report that Pkd2l1/C21orf59/Kurly, a cytoplasmic protein enriched in the base of cilia and needed for their motility, is required for cerebrospinal fluid (CSF) flow detection and for spine straight.

Using zebrafish as model organism, they monitor the dynamics of motile cilia as well as that of endogenous and exogenous particles in the central canal and report on a bidirectional movement in the spinal cord (i.e.: anterior to posterior ventrally, and posterior to anterior dorsally). They assess the CSF flow in the Kurly mutant fish upon injection of fluorescent beads in the hindbrain and found that they remained confined to the brain and in some cases, or to the rostral most portion of the central canal.

They postulated that CSF contacting neurons (CSF-cNs) must be the cells that sense and monitor changes in the CSF. They then performed a series of experiments that indicate that Pkd2l1/C21orf59/Kurly is expressed in CSF-cNs and that mutations in this gene affect the physiological properties of these neurons as assessed by calcium imaging and channels

dynamics.

Finally, the authors used alizarin red that stains bony skeleton and report on an abnormal curvature of the spine in *pkd211* mutant fish.

Although presented as novel findings, many pieces of data have already been published (see below) and the new ones are too preliminary and "suffer" many imperfections/inconsistencies.

Major concerns:

1. Defects in cilia motility/function and in body curvature have already been described in zebrafish with mutations in the *Pkd211* gene (Jaffe et al., 2016) and should not be presented as new findings.
2. The bidirectional flow in the central canal might be interesting. However:
 - a) The movies that would support this claim are missing and the reader has to trust images tracing the movement of beads
 - b) The flow is preserved in the *pkd21-/-* mutant fish. So how can CSF-cNs, which form the lining of the central canal, detect defects in a normal flow and respond differentially to similar SCF flows?
3. The authors used different mutant lines in the study (e.g. *Kurlytm302* for CSF flow dynamics, *pkd21-/-* for validation of the *pkd211* antibodies and electrophysiology of CSF-cNs, *Kurly* morphants to show that the sensory properties of CSF-cNs are not affected by the *Kurly* mutations). This is extremely disturbing and makes the interpretation of the results almost impossible.
 - a) What is the rationale behind this?
 - b) To what extent *Kurlytm302/ tm302/* and *pkd21-/-* are similar or different?
 - c) Why using morpholinos while different mutants are available?
 - d) In a previous study by the same group (Böhm et al, 2015), the CSF-cNs response to body/tail deformities is abolished in *pkd21-/-* mutants. How can the authors ascertain that the impaired calcium activity in *Kurlytm302/ tm302/* is not secondary to body deformities? Is it different from what they have reported in *pkd21-/-* mutants?

RESPONSE TO REVIEWERS

We thank the reviewers for their valuable feedback.

Please note that official nomenclature for the *kurly* gene used in this study has been changed to *cfap298* for cilia and flagella associated protein 298. The mutant used in this study previously identified as *kurly^{tm304}* is now called *cfap298^{tm304}*. We have updated the name throughout the text and use the official nomenclature below.

All edits to the main text appear in red in the revised manuscript.

Reviewer #2

1. The defect of spinal curvature in the Kurly mutant looks more serious than that in the Pkd2/1 mutant. Therefore, I think the authors should measure and compare the spinal curvature defects in both the Kurly and Pkd2/1 mutant to clarify the importance of Pkd2/1 channels in spinal curvature.

Adult cfap298^{tm304} and pkd2l1 mutants

The defect of spinal curvature in adult *cfap298^{tm304}* mutants (Grimes *et al.*, 2016) is indeed more severe than that in adult *pkd2l1^{icm02/icm02}* mutants described in our study.

cfap298 mutants form a torsional scoliotic spine seen as early as few weeks of development in zebrafish juveniles (Grimes *et al.*, 2016). Scoliosis is a three-dimensional deformative abnormality of the spine defined by the Cobb angle of spine curvature in the coronal plane, and accompanied by vertebral rotation in the transverse plane and hypokyphosis in the sagittal plane. The rotation component starts when the scoliosis becomes more pronounced.

In contrast, adult *pkd2l1^{icm02/icm02}* mutants only display 2D abnormality of the spine curvature occurring as kyphosis at the level of the precaudal spine where the rib cage is located (Fig. 4 and Supplementary Fig. 3).

We have proposed that Pkd2l1 contributes to maintenance of proper 2D spine morphology but emphasized that Pkd2l1 does not cause torsion in 3D clinically associated with idiopathic scoliosis. We have modified the text to clarify that the defects of the *cfap298^{tm304}* adult fish, which is similar to an idiopathic scoliosis phenotype, versus the defects in adult *pkd2l1^{icm02/icm02}* mutant fish that present with kyphosis as described above.

Mutant cfap298^{tm304} and pkd2l1 embryos

The above description applies for adult stages. At the embryonic stage, *cfap298^{tm304}* zebrafish will develop a curled-down body axis if the embryos are maintained at the standard 28°C and die at larval stages, presumably because they are unable to properly swim, and thus feed. However, if the embryos are raised at a lower temperature (25°C), they will have normal spine curvature, as previously described in Jaffe *et al.*, Cell Reports 2016 and Grimes *et al.*, Science 2016. If these animals are raised at 25°C until juvenile stages and then undergo a temperature change to 28°C, then they will develop the scoliotic phenotype, surviving until adulthood. *pkd2l1^{icm02/icm02}* mutant embryos do not have any body axis defects, now shown in Supplemental Fig. 3a-c, and their curvature phenotype only starts to become evident at around 5 months of age.

RESPONSE TO REVIEWERS

2. In the study, the expression of Pkd2/1 channels was showed in the apical extension of CSF-cNs in the trunk. Is there any expression of Pkd2/1 channels in the brain. It would be nice to exclude the possibility that spinal curvature defects in Pkd2/1 mutants are not due to the disruption of the activity of neurons in the brain. Therefore, I suggest the authors could express a neurotoxin, like the BoTxBLC, specifically in CSF-cNs to demonstrate their importance in spinal curvature defects.

Concerning the expression of the Pkd2/1 channel in the brain

We previously reported no expression in the brain of the *pkd2/1* gene (Djenoune et al., Frontiers in Neuroanatomy 2014). Recently, the Lewis Lab performed an extensive characterization of the entire *pkd* gene family in zebrafish from embryonic to larval stages and confirmed our observations (England et al., Front. Cell Dev. Biol 2017). *pkd2/1* is localized to CSF-cNs and taste receptors and no *pkd2/1* expression was reported in brain structures. Furthermore, immunohistochemistry for the Pkd2/1 protein performed at 1 dpf and 3 dpf in our lab confirmed these observations (Cantaut Belarif and Wyart, *unpublished observations*).

Relevance of spinal defects

We attempted to raise some *Tg(pkd2/1:gal4; UAS:BoTxBLC-GFP)* to adulthood, however these fish had high mortality rates. Because the surviving fish grew at different rates compared to their control non-expressing siblings, we could not compare the phenotype in these animals.

However, due to the specificity of the expression of *pkd2/1* in CSF-cNs detailed above, we believe the *pkd2/1* mutant informs on the role of CSF-cNs during development. In a third generation of animals, we reproduced the phenotype observed (kyphosis) in *pkd2/1* mutants in compared to their control siblings (Supplementary Fig. 3).

3. In the fig.S2, the mutant rather than morphant of Kurly should be examined to demonstrate the integrity of CSF-cNs' ability for responding to mechanical stimuli.

We agree entirely and have confirmed these results with the *cfap298^{tm304}* mutant (see Supplemental Figure 4). During spontaneous contractions of the embryo, *cfap298^{tm304}* still responded, though this response was decreased compared to wild type, comparable to what we previously showed with the *cfap298* morpholino.

Reviewer #3:

1. Defects in cilia motility/function and in body curvature have already been described in zebrafish with mutations in the Pkd2/1 gene (Jaffe et al., 2016) and should not presented as new findings.

There is a confusion here, probably due to our writing, which we clarified in the new version of the manuscript. Cfap298 is a ciliary protein required to recruit the outer dynein arms necessary for the cilia to beat (Jaffe et al., 2016). In contrast, Polycystic Kidney Disease 2-like-1 (Pkd2/1) is a transient receptor potential channel present in sour taste receptors (Huang et al., 2006) and previously reported to be involved in detecting changes

RESPONSE TO REVIEWERS

in pH and osmolarity CSF-contacting neurons (Orts-del'Immagine *et al.*, 2014, 2016; Böhm, Prendergast *et al.*, 2016; Jalalvand *et al.*, 2016).

Previously, Jaffe *et al.*, 2016 reported effects of the *cfap298* mutation on ciliary function as well as the role of Cfap298 in planar cell polarity in zebrafish. Soon after, Grimes *et al.*, 2016 reported the mutations that led to defects in ciliary motility, including defects in the *cfap298* gene, lead to spinal curvature defects. Note that none of these publications investigate the role of the Pkd2l1 channel in spine morphogenesis.

In contrast, in our manuscript, we report that the transient receptor potential (TRP) ion channel previously involved in sensory signaling for pH detection, Polycystic Kidney Disease 2-like-1 is a mechanoreceptor leads to spinal curvature defects. This finding is new; it was not investigated in previous studies that focused on pH (Orts-Del'Immagine *et al.*, 2016) or ASIC channels (Jalalvand *et al.*, 2016).

This ion channel is different from Cfap298, which is a ciliary protein required to recruit the outer dynein arms necessary for the cilia to beat. We have edited the text to clarify the differences between these two different genes and proteins. We apologize for the confusion.

2. The bidirectional flow in the central canal might be interesting. However:

a) The movies that would support this claim are missing and the reader has to trust images tracing the movement of beads

We apologize that the movies were not attached to the manuscript when the manuscript was transferred to Nature Communications. The accompanying 5 movies have been added to this version of the text, which include:

Movie S1. Cilia in a 24 hpf *Tg(β-actin:Arl13-GFP)* embryo. Images were acquired at 33 Hz and are played in real time.

Movie S2. Imaging of exogenous beads in the central canal of the spinal cord in a wild type embryo. Images were acquired at 10 Hz and are played in real time.

Movie S3. Imaging of endogenous particles with FF-OCT in the central canal of the spinal cord in a 26-28 hpf *Tg(β-actin:Arl13-GFP)* embryo. Images were acquired at 100 Hz and are played in real time.

Movie S4. Calcium imaging of CSF-cNs in a wild type *Tg(pkd2l1:GCaMP5G)* embryo at 24-26 hpf. Images were acquired at 4 Hz and are sped up 5x in the movie.

Movie S5. Calcium imaging in CSF-cNs in a *cfap298^{m304/tm304}* *Tg(pkd2l1:GCaMP5G)* embryos at 24-26 hpf. Images were acquired at 4 Hz and are sped up 5x in the movie.

Movie S6. Calcium imaging of CSF-cNs in a *pkd2l1^{-/-}* *Tg(pkd2l1:GCaMP5G)* embryo at 24-26 hpf. Images were acquired at 4 Hz and are sped up 5x in the movie.

b) The flow is preserved in the *pkd21^{-/-}* mutant fish. So how can CSF-cNs, which form the lining of the central canal, detect defects in a normal flow and respond differentially to similar SCF flows?

Our findings indicate that activity in wild type embryos is driven by Pkd2l1. In contrast, *pkd2l1^{icm02/icm02}* embryos no longer show activity, possibly because the sensory Pkd2l1 channel itself is required for mechanosensation and detection of cerebrospinal fluid flow.

RESPONSE TO REVIEWERS

Again, we would like to reiterate that Pkd2l1 is not a ciliary protein required for motility, but a transient receptor potential channel that transduces sensory input. The Pkd2l1 channel is located in the apical extension of cells in contact with the CSF, which includes a cilium but also a dense brush of microvilli, on which the Pkd2l1 channel is located (see microvilli labeled in 3a-b, Supplementary Fig. 5).

In vivo activity observed in CSF-cNs at embryonic stages recorded with calcium imaging (Fig. 2 and Fig. 3) or with electrophysiology (Fig. 3) requires the Pkd2l1 channel. These *in vivo* observations, combined with *in vitro* recordings of wild type and *pkd2l1^{icm02/icm02}* CSF-cNs during mechanical stimulation suggest that the Pkd2l1 channel is necessary for spontaneous activity and mechanical integration in CSF-cNs.

All together, we demonstrate *in vitro* for the first time on isolated CSF-cNs that these cells are direct mechanosensors, and that their mechanosensitivity requires the Pkd2l1 channel, as it is abolished in *pkd2l1^{icm02/icm02}* mutants. In *pkd2l1^{icm02/icm02}* mutants, CSF flow is normal because ciliary motility is not disturbed, however, the CSF-cNs are no longer mechanosensory because these cells lack the channel, Pkd2l1, required to detect movement (bending of the tail, deflection of CSF-cN membrane or CSF flow).

3. The authors used different mutant lines in the study (e.g. Kurly^{tm302} for CSF flow dynamics, *pkd21^{-/-}* for validation of the *pkd2l1* antibodies and electrophysiology of CSF-cNs, Kurly morphants to show that the sensory properties of CSF-cNs are not affected by the Kurly mutations). This is extremely disturbing and makes the interpretation of the results almost impossible.

a) What is the rationale behind this?

b) To what extent Kurly^{tm302}/ ^{tm302} and *pkd21^{-/-}* are similar or different?

We acknowledge that the rationale for using each mutant was not explicit enough. We apologize for the confusion with the different mutant lines and have edited the text to clarify our intentions. We used two mutants in this study, described here for clarity:

1/ Kurly/Cfap298 is a ciliary protein expressed in a wide diversity of cells with motile cilia and is required for planar cell polarity and recruitment of the outer dynein arms of cilia needed for ciliary motility (Jaffe *et al.*, 2016). In *cfap298^{tm304}* mutants, ciliary motility is lost, and flow of the CSF is disrupted in central canal of the spinal cord.

2/ Pkd2l1 is a transient receptor potential channel, expressed in the apical extension (primarily microvilli, see Fig. 3a-b and Supplementary Fig. 5) of CSF-cNs, which is required for mechanosensation. We observed that Pkd2l1 is specific to CSF-cNs in the spinal cord (Djenoune *et al.*, 2014). Pkd2l1 is necessary for sensory function (this manuscript, Böhm *et al.*, 2016) but is not required for ciliary motility, as we showed in a previous study that cilia motility was not affected in *pkd2l1^{icm02/icm02}* (Böhm *et al.*, 2016).

Our rationale for using these two mutants was to first demonstrate that CSF-cN activity is a readout of CSF flow in the central canal. We show here that CSF-cN activity requires the channel Pkd2l1 and the motility of cilia, as a loss of ciliary motility and flow defects in the *cfap298^{tm304}* mutant lead to defects in CSF-cN signaling.

Second, we wanted to identify the mechanism by which CSF-cNs can detect flow in the central canal. Because Pkd2l1 is implicated in sensory function in CSF-cNs, we wanted to determine whether loss of this channel might lead to a defect in detection of CSF flow.

RESPONSE TO REVIEWERS

pkd21^{icm02/icm02} mutants no longer have calcium or electrical activity, even though these mutants have normal CSF flow, implicating Pkd211 in generating CSF-cN activity, putatively reflecting flow in the central canal. As Pkd211 is not thought to be required for ciliary motility or flow, it was not surprising that flow was normal in these mutants. We demonstrated that Pkd211's role in mechanosensation with *in vitro* recordings of wild type and *pkd21^{icm02/icm02}* CSF-cNs.

c) Why using morpholinos while different mutant are available?

We agree that use of both a morphant and a mutant could confound interpretation of results and have replaced the figure using the *cfap298* morphant with data acquired in the *cfap298^{tm304}* mutant, which has similar results showing that the cells still respond, albeit with a smaller amplitude (Supplementary Fig. 4).

d) In a previous study by the same group (Böhm et al, 2015), the CSF-cNs response to body/tail deformities is abolished in *pkd21^{-/-}* mutants. How can the authors ascertain that the impaired calcium activity in *Kurlytm302/ tm302/* is not secondary to body deformities? Is it different from what they have reported in *pkd21^{-/-}* mutants?

In Böhm et al, 2016, we demonstrated that acute passive (push) or active (during locomotion) bending of the tail leads to activity in CSF-cNs in a *pkd211*-dependent manner. One could indeed ask whether the reduced activity of CSF-cNs in *cfap298* mutants with a curled down phenotype is not secondary to defects in body axis formation.

Since the response to bending of CSF-cNs is transient when we apply pressure for long durations (Bohm & Wyart, unpublished observations), we do not believe that this could be the case. In addition, *cfap298^{tm304/tm304}* mutant embryos spontaneously twitch as regular wild type animals. When they do so, the response to spontaneous contractions is reduced in the *cfap298^{tm304/tm304}* mutants compared to their wild type and heterozygote siblings.

Since we cannot exclude that the deformed body axis could contribute to the change in activity levels of CSF-cNs, we added a sentence in the revised manuscript (line 167).

REVIEWERS' COMMENTS:

Reviewer #2 (Remarks to the Author):

My comments have been well addressed in the revised manuscript, and the study is much improved. I have no more comment now.

Reviewer #3 (Remarks to the Author):

The authors have adequately addressed most of the concerns and I am happy to support the publication of the paper.